# The Effect of Acute Exercise on State Anxiety: A Systematic Review

**DOI:** 10.3390/sports11080145

**Published:** 2023-08-01

**Authors:** Madeleine Connor, Elaine A. Hargreaves, Orla K. Scanlon, Olivia K. Harrison

**Affiliations:** 1School of Physical Education, Sport and Exercise Sciences, University of Otago, Dunedin 9016, New Zealand; elaine.hargreaves@otago.ac.nz (E.A.H.);; 2Department of Psychology, University of Otago, Dunedin 9016, New Zealand; olivia.harrison@otago.ac.nz; 3Nuffield Department of Clinical Neurosciences, University of Oxford, Oxford OX3 9DU, UK; 4Translational Neuromodeling Unit, University of Zurich and ETH Zurich, 8001 Zurich, Switzerland

**Keywords:** physical activity, psychological well-being, stress, tension, aerobic

## Abstract

Acute exercise has been shown to induce a small reduction in state anxiety, yet the most beneficial exercise stimulus is not clear. This review provides an update on the papers published since the last comprehensive review in 2015, with specific emphasis on whether study quality has improved. Randomised control trials, conducted in samples of healthy adults with non-clinical anxiety, were sourced from PubMed, PsycInfo, and Scopus. Study characteristics and study quality were assessed in nine studies comprising thirteen exercise conditions. Acute exercise significantly reduced anxiety in 53% (*N* = 7/13) of the exercise conditions. In comparison to a control condition, four showed exercising to be more effective, and one was as effective as the control. Two of the effective studies did not contain a control group. Six conditions were ineffective in reducing anxiety. There was no clear pattern of what combination of exercise mode, duration, and intensity was most effective, suggesting a variety may be effective in reducing anxiety. Methodological limitations still exist within the research, e.g., participant recruitment not considering baseline anxiety; variations in the control condition content. Future research should include participant samples exhibiting moderate-to-high levels of anxiety and examine self-selected exercise intensities.

## 1. Introduction

In New Zealand, 16% of adults reported experiencing moderate levels of state anxiety in 2020 [1], and there has been a significant increase in the numbers accessing mental health care from 2016/2017 to 2021/2022 [2]. While low levels of state anxiety may be functional, protecting us from potential threats [3], above-average (while not necessarily disordered) anxiety has the potential to interfere with daily functioning [4,5]. Elevated anxiety is associated with an increased propensity for anxiety to progress to clinical levels [4] and an increased risk of cardiovascular disease [6]. A meta-analysis of prospective cohort studies has shown that those who engage in higher levels of physical activity have a lower risk of developing anxiety symptoms [7]. Further, acute exercise has the potential to lower state anxiety (For review, see: [8,9]), providing an accessible strategy for individuals to manage their day-to-day experiences of anxiety. Given the rise in those experiencing anxiety, this review provides an update to the literature examining the use of acute exercise to reduce state anxiety and the effects of the specific exercise stimulus used, aiming to provide guidance for the provision of exercise to manage elevated state anxiety.

There have been two previous meta-analyses examining the effects of an acute bout of exercise on state anxiety [8,9]. Petruzzello et al. [8] provided the seminal work, which was extended by Ensari et al. [9] nearly 25 years later in 2015. As noted by Ensari et al. [9], both reviews reported a small but statistically significant effect of acute exercise in reducing anxiety symptoms when compared to a control condition. However, both studies also identified methodological limitations within the literature, meaning the potential true effect of exercise could not be ascertained. Ensari et al. [9] assessed the study quality using the Physiotherapy Evidence Database (PEDro) scale [10], with a score of six or greater indicating a high study quality [10]. Only 12 of 36 studies (33%) met this threshold of six or higher. Pertinently, the PEDro scores were found to moderate the acute effects of exercise on anxiety, with studies of higher quality associated with greater reductions in anxiety. Methodological limitations included the continued reliance on convenience sampling without strict inclusion criteria. Resultantly, participants had low levels of anxiety at the measured baseline, creating floor effects in the analysis (e.g., [11]). Further reducing the study quality was the lack of blinding. While the blinding of participants and therapists (i.e., those delivering the exercise) is methodologically difficult, no studies were successful in blinding the assessors either. The barriers to blinding participants and therapists may highlight the greater importance of working with blinded assessors, and ensuring that other aspects of study quality are robust. Ensari et al. [9] concluded that, in order to accurately assess the anxiolytic effect of acute exercise, study rigour must be improved. Thus, it is timely to review the recent literature to evaluate whether this conclusion has been heeded and if recent research has been conducted with more rigorous methodologies. This would enable a more thorough understanding of the acute exercise–state anxiety relationship, and establish a precedent for future protocols.

Understanding the exercise stimulus (mode, duration, and intensity) required to improve anxiety is important to ensure that the exercise that people engage in will be effective in reducing anxiety [12]. In their review, Ensari et al. [9] concluded that higher intensities and treadmill modalities were associated with greater reductions in anxiety compared to other intensities or alternate aerobic and muscle-strengthening modalities; however, they also noted there were too small a number of studies investigating some of the exercise factors analysed to make strong conclusions. Therefore, the purposes of this study were to (1) examine the effects of an acute bout of exercise on state anxiety from the recent literature while investigating the quality of recently published research and (2) to review the characteristics of the exercise stimuli used within the studies to ascertain whether there is an ‘optimal’ bout of exercise that can be prescribed to benefit state anxiety.

## 2. Materials and Methods

### 2.1. Search Strategy

A literature search was conducted of PubMed, Scopus, and PsycInfo in January 2023. Databases were searched using the following search strategy: (“Exercise” OR “Physical Activity”) AND (“Anxiety” OR “Anxiety Sensitivity”) AND “Intervention”. The results were limited to humans, adults, and articles published in English. The results were also limited to those published after 2015 because our focus was to expand on the meta-analytic findings of Ensari et al. [9].

### 2.2. Eligibility Criteria

Eligible studies were experimental trials that (1) investigated the effect of a structured acute bout of exercise, compared to another exercise stimulus or a control condition, on state anxiety; (2) contained anxiety as a primary outcome; (3) included adults aged 18 to 45 years old who did not suffer from a diagnosed medical disorder. The upper age limit of 45 years was set to reduce the likelihood that women undergoing menopause would be included, as menopause has been shown to influence anxiety [13,14].

### 2.3. Study Selection

The study titles were screened for relevance by two reviewers (MC and OS). Titles were most often excluded due to the population examined, a lack of exercise, or anxiety not being a primary outcome. The full text was then retrieved for those deemed potentially relevant, and the abstracts were reviewed for eligibility to provide the final studies for analysis.

### 2.4. Analysis

From the studies deemed to have met the eligibility criteria, study information and data regarding the participants, the exercise conditions, anxiety measurements, additional outcome measures, study results and associated effect sizes (Cohens *d* was used for the parametric group comparisons, the rank-biserial correlation was used for the non-parametric group comparisons, and η^2^ was used for the ANOVA analyses) were extracted. The effect sizes were interpreted according to the benchmarks reported in Lakens [15]. Conditions were deemed to be effective equal to the control condition, effective superior to the control condition, or ineffective, and were further categorised according to the exercise parameters for analysis (see Appendix A, Table A1). Study quality was assessed using the PEDro scale [10], which comprises 11 true or false items regarding the methodological study qualities (e.g., subjects were randomly allocated to groups). Points were awarded if the study had employed the specific methodological criteria contained in the item. If it was unclear as to whether or not the criteria were included, the point was not awarded. For studies with a cross-over design, points towards the items for group randomisation and concealment of allocation (i.e., when researchers communicated with a prospective participant to determine eligibility, they were unaware of which condition the person would be allocated to) were awarded if the group allocation was concealed, and/or randomised which condition the participant would complete first. The points awarded to each study were summed (excluding the first item, in line with instrument instructions [10]), providing a maximum total score of 10. A cut-off score of 6 was used to indicate ‘good’ study quality in line with previous studies (e.g., [9]) and instrument instructions [10].

## 3. Results

### 3.1. Study Selection

As detailed in Figure 1, the search strategy provided 3608 studies, which were reduced to 3423 once the duplicates were removed. The titles of those studies were screened, and 42 studies were deemed relevant. The abstracts of these 42 studies were examined for eligibility, and 9 studies, comprising 13 exercise stimuli, met the eligibility criteria and were subject to review.

### 3.2. Study Characteristics

Table 1 summarises the characteristics of the eligible studies. Thirteen exercise stimuli were contained within the nine included studies. In these nine studies, the sample size ranged from 17 [16] to 88 [17] (M = 51.33, SD = 25.04). Five studies employed an inactive control condition [17,18,19,20,21], while one study included both an inactive control and an alternative treatment condition (biofeedback breathing [22]). One study included a low-intensity exercise condition (cycling at 10–20% HR reserve [16]), which the authors considered an active control condition due to the very light intensity. In the studies with control conditions, two also employed comparison exercise conditions [19,21]. Finally, two studies had comparison exercise conditions without inactive control conditions [23,24].

Seven studies assessed anxiety using the State-Trait Anxiety Inventory (STAI; [25]) [16,17,20,21,22,23,24], while two utilised the Anxiety Sensitivity Index-3 (ASI-3; [26]) [18,19]. Three studies specified a minimum anxiety level in their inclusion criteria [17,20,22]: Herzog et al. [17] stipulated a high ASI-3 [26] score of ≥17; Cooper and Tomporowski [20] categorised participants according to their STAI-T [25] score as low (≤33) or high (≥39) anxiety; and Meier and Welch’s [22] inclusion criteria stipulated an above average (≥14.2) perceived stress score [22,27].

Overall, 53% (*N* = 7) of the 13 acute exercise conditions resulted in significantly reduced anxiety from pre- to post-exercise [17,18,19,20,23]. Of those seven, four resulted in a significantly greater reduction in anxiety compared to a control condition ([18,19,20] shown in Table 1). Two of the exercise conditions were contained within one study, without a control employed, and were both found to be effective, with no difference in efficacy between the conditions [23]. Finally, one exercise condition was as effective as the non-active control condition (documentary viewing, [17]).

Of the six exercise conditions that did not result in significant pre- to post-exercise reductions in anxiety, two were not compared to a control condition [24]. Two had similar non-significant effects to an inactive control condition (quiet rest, [21]), and one condition had similar non-significant effects as its active control (low intensity exercise, [16]). One exercise condition, and its inactive control (studying), both had non-significant effects on anxiety. However, the alternative treatment condition contained within this study was effective in reducing anxiety ([22], biofeedback task).

### 3.3. Role of Modality

Twelve conditions utilised an aerobic exercise modality, and one comprised muscle-strengthening exercise. Of the twelve aerobic-based stimuli, six employed cycling [16,17,18,19,20], three utilised a treadmill [21,22,23], and three involved exergames (the combining of exercise with video games [28]) [23,24].

Of the six cycling conditions, four resulted in significantly greater anxiety reductions than the control group [18,19,20], with small [19] to medium [18,20] effect sizes. One was as effective as the control condition (with insufficient data to estimate the effect size) [17], and one had similar non-significant effects to the control condition [16] (with insufficient data to estimate the effect size). In the treadmill-based conditions, two were not significantly different from the control [21,22]; one was as effective (large effect size) as the comparison exercise condition contained within the same study (exergame, [23]).

When exergames were used, one was as effective as the comparison (treadmill [23]), while the remaining two contained within one study were as effective as each other (single- vs. multi-player mode [24]) with trivial and medium effect sizes between modes, respectively.

The one study that contained a muscle-strengthening condition [21] did not report any pre- to post-reductions in anxiety. The condition was compared to 20 min of moderate intensity treadmill-based exercise and quiet rest. No conditions within this study, including the quiet rest control, were found to be effective. The muscle-strengthening protocol involved 20 min of various resistance exercises. The intensity was poorly quantified, with an exercising heart rate reported to be between M = 106.5–133 BPM (rather than as a percentage of the maximal heart rate), depending on the movement completed.

### 3.4. Role of Intensity

Five of the twelve (41%) aerobic-based stimuli used vigorous intensity exercise (defined as 71–90% of the maximum HR/V̇O_2_, or 61–85%HR_reserve_ [29]) [16,18,19,23]. Four of those stimuli (80%) resulted in a significant decrease in anxiety pre- to post-exercise [18,19,23]. Of these four, two were contained within one study and were as effective as each other (treadmill vs. intensity-matched exergame [23]) with a large effect size of time across both groups. Two were more effective than the control group (stretching, [18]; and waitlist, [19], with small [19] and medium [18] effect sizes. The remaining condition was ineffective, as was the control group (low intensity exercise, [16]).

Moderate intensities (55–70% HR/V̇O_2 max_, or 40–60% HR_reserve_ [29]) were investigated in four of the twelve stimuli [17,19,20,21], with three of those four studies (75%) showing significantly reduced anxiety pre- to post-exercise [17,19,20], with small [19] to medium [20] effect sizes (one study had insufficient data to estimate the effect size [17]). Two reported significantly greater reductions than the control (waitlist, [19], and quiet rest, [20]), and one reported similar reductions to the control condition (documentary watching, [17]). The remaining condition was ineffective, as was the control condition (quiet rest) [21].

Intensity was not quantified appropriately in two studies [22,24]. In one, the HR was provided in beats per minute [24], and in the other, the intensity was described as self-paced and no measure of intensity was given [22]. In both of these studies, the exercise stimuli did not result in significant anxiety reductions, despite the authors reporting small (single-player mode-[24]) to medium effect sizes (multi-player mode-[22,24]).

### 3.5. Role of Duration

Exercise duration was categorised as either less than, equal to, or greater than 30 min for the purposes of analysis. This was in consideration of the World Health Organisation recommendations of 30 min of exercise per day, 5 days per week [30]. Four out of thirteen exercise conditions (33%) had durations greater than 30 min, all lasting 45 min [19,23]. The results showed that all of these stimuli resulted in significant reductions in anxiety with small [19] and large [23] effect sizes. Two stimuli showed significantly greater reductions than the control condition (waitlist, [19]), and two exercise stimuli showed equal reductions (treadmill vs. intensity matched exergame, [23]).

Four conditions had durations of less than 30 min [18,20,21,22], ranging from 10 min [22] to 25 min [20]. Only two conditions showed significant reductions in anxiety (with durations of 20 [18] and 25 [20] min), with both showing greater reductions compared to the control (stretching and quiet rest, respectively) and medium effect sizes. One condition [21] (20 min in duration) failed to reduce anxiety, although no reduction was found in the comparison (resistance training) or control (quiet rest) condition either. The remaining condition of 10 min duration [22] and its control group (studying) failed to reduce anxiety.

Finally, four conditions employed a 30 min duration [16,17,24], but only one resulted in significant reductions in anxiety [17] with no significant difference to the control condition and insufficient data to estimate the effect size. No reduction occurred in two stimuli (no control group-[24]). The remaining stimulus and control condition (low intensity exercise), did not influence anxiety [16] (insufficient data to estimate the effect size).

### 3.6. Evaluation of Study Quality

The evaluation of study quality is shown in Table 2. Five of the nine studies reviewed (55%) obtained a PEDro score of six or greater [18,19,20,21,24], indicating high study quality [10]. Two studies scored five [17,23], and two scored four [22,31], indicating lower study qualities. All studies were awarded a point for specifying eligibility criteria (item 1), obtaining measurements from more than 85% of participants (item 8), reporting between group results (item 10), and providing time point measurements and measurements of variability (item 11). Only five were awarded points for concealing the allocation at randomisation (item 3), and six were awarded a point for similarity at baseline (item 4). As discussed, no points were awarded for items 5 through 7 regarding blinding.

### 3.7. Control Group

A variety of control conditions were employed across the different studies. Seven of the nine studies contained a control group [16,17,18,19,20,21,22]. Of these, five were ‘inactive’, e.g., waitlists or quiet rest [17,18,19,20,21]; and two were ‘active’ (10–20%HR reserve, identified as a control condition by the authors-[16], stretching-[18]). Of the inactive controls, one (documentary watching-[17]) was found to also decrease anxiety. Two studies found that exercise was more effective than the inactive control (waitlist-[19], quiet rest-[20]), while another found that no condition, including the control, reduced anxiety (quiet rest-[21]). One study [22] compared exercise to an inactive control (studying) and an alternative treatment (biofeedback), finding that only the biofeedback reduced anxiety. Of the two studies which included an active control group [16,18], neither control condition resulted in lower anxiety. Of all the control conditions, only one reduced anxiety (documentary watching-[17]), although to the same extent as the exercise condition.

## 4. Discussion

This review aimed to examine the effects of an acute bout of exercise on state anxiety, updating the literature since the last review was completed in 2015 [9]. Importantly, this review examined whether the recent research had heeded the conclusions of Ensari and colleagues and had been conducted with more rigorous methods to enable the true effect of exercise to be ascertained. We also focused on the composition of different acute exercise bouts to identify whether there is an ‘optimal’ stimulus to reduce state anxiety. Nine studies comprising thirteen exercise conditions were investigated. There were relatively fewer studies investigated than the 36 studies of Ensari et al. [9]. However, the present study excluded clinical groups and investigated a temporal space of only 8 years, compared to approximately 24 years. Of the 13 exercise conditions we investigated, 12 utilised an aerobic modality, 5 utilised a vigorous intensity, and durations varied evenly across the >30, =30, and <30 min categories. Positively, 53% of the exercise conditions investigated resulted in reduced anxiety, with the effect sizes ranging from small (*d* = 0.35 [19]) to large (*d* = 1.29 [23]). These results suggest that exercise has the potential to significantly reduce anxiety with a range of effect sizes, complementing the findings of Ensari et al. [9] and Petruzzello et al. [8]. Despite the relative improvement in study quality over the last eight years, our results show that many of the methodological limitations that were identified by Ensari et al. [9] are still present. Thus, the ability to draw definitive conclusions on the acute effect of exercise on anxiety from the literature is still limited. Finally, there were few evident patterns regarding the most optimal exercise stimulus to reduce anxiety. Generally, studies employing moderate and vigorous intensity of longer durations showed significant pre- to post-exercise differences, with varied effect sizes. However, evidence is lacking to evaluate the effects of self-selected intensities and consideration of muscle-strengthening modalities.

Aerobic forms of exercise are clearly preferred over muscle-strengthening exercises in these lab-based studies. This may be due to their potential to better target the hypothesised physiological drivers of the anxiolytic effect of exercise, such as the monoamine [32] or respiratory systems [33]. Recent investigations of chronic muscle-strengthening exercise have highlighted its potential to reduce anxiety [34]. However, it has also been noted that there is a lack of well-designed investigations, particularly in comparison to aerobic exercise [35], limiting the ability to draw conclusions regarding their efficacy in an acute exercise context. The relative number of studies investigating muscle-strengthening modalities has also decreased since Ensari et al. [9]—of the 36 stimuli investigated by Ensari, 16 (44%) utilised muscle-strengthening modalities compared to the 7% (*N* = 1/13) in the present review. Ensari et al. [9] also noted that the lack of representation of muscle-strengthening modalities within the literature limits the conclusions that can be drawn.

Durations of less than, equal to, and greater than 30 min have been employed equally as often, and the results showed that durations of 45 min consistently showed reductions in anxiety. Only three of eight stimuli with durations of 30 min or less were effective. In their 1991 review, Petruzzello et al. [8] concluded that duration was a key variable in the anxiolytic effect of exercise. They reported durations greater than 20 min to be more effective, somewhat reflecting the current finding where longer intensities were more effective. However, research that compares the effect of exercise of different durations within a single study is still required to confirm an optimal duration—a sentiment that was first noted in 1991 by Petruzzello et al. [8].

The majority of conditions (*N* = 9/13, 69%) used either a moderate or vigorous intensity of exercise and all but two resulted in significant improvements in anxiety. Effective studies employing these intensities displayed small (*d* = −0.35 [19]) to large (*d* = 1.29 [23]) effect sizes. This again reflects the findings of Ensari et al. [9], who noted that high exercise intensities might be more effective, potentially due to the hypothesised contribution of anxiety sensitivity to the anxiolytic effect of anxiety [18,36]. Exposure to anxiety-related symptoms, such as an elevated heart or breathing rate via exercise, may provide a setting for individuals to habituate to and reappraise such symptoms, reducing anxiety sensitivity [37] and, therefore, anxiety. Higher intensities have been shown to improve anxiety sensitivity more rapidly than low intensities [38], possibly due to their generation of a greater perturbation to physiological systems.

Notably, a high-quality investigation of the acute effect of self-selected exercise intensities on anxiety has not been conducted. Meier and Welch [22] allowed participants to self-select their intensity and compared this condition to biofeedback and a control. Their results showed neither of their conditions had a significant effect on anxiety. Interestingly, the participants in this study only reported an RPE of 9, indicative of a ‘very light’ intensity [39], which may explain the lack of effect. Other studies investigating the affective responses to exercise have shown that non-anxious individuals select exercise at moderate or high intensities on average (e.g., [40,41]). Meier and Welch [22] did not provide the instructions that were given to their participants on how to self-select their intensity, and thus these instructions may explain the differences from the other studies. Alternatively, perhaps anxious individuals prefer to exercise at lower intensities, although those with clinical anxiety and/or depression have previously selected to exercise at approximately 77% of their age-predicted maximum heart rate [42]. To better understand the effect of self-selected intensities, well-designed studies should be conducted. A thorough investigation of self-selected intensities may indicate the benefits of allowing individuals to manage their own exertion, enhancing psychological constructs such as self-efficacy [41] and affect [43] alongside the study’s ecological validity.

When considering intensity, duration, and modality, the studies examined alongside previous research do not present a clear optimal exercise stimulus to reduce anxiety. It may be of further benefit to consider the overall volume of exercise. For example, while LeBouthillier and Asmundson [18] and Broman-Fulks et al. [21] both comprise 20-minute durations, they each have different intensities, quantified according to different measures. This may then contribute to why LeBouthillier and Asmundson [18]’s higher intensity sprint intervals at 60–80% age-adjusted heart rate reserve resulted in a significant reduction in anxiety, while no such reduction was found from Broman-Fulks et al. [21]’s continuous protocol at 65–75% of participants’ maximum heart rate. However, the volume of exercise cannot be directly compared across studies due to the differences in the descriptions of intensity, as well as the inadequate quantifications of the intensity undulations within an acute bout. Further research should provide a more precise measurement of the exercise dose (such as energy expenditure) to allow for consideration of the dose–response relationship, as demonstrated within the depression literature (e.g., [44]).

Overall, study quality has improved since the review conducted by Ensari et al. [9]. Indeed, 55% of the studies reviewed in this study can be classed as high quality, whereas only 33% of the 36 studies reviewed by Ensari et al. [9] scored higher than 6. Despite this, some of the methodological limitations reported by Ensari et al. [9] remain. Positively, all studies provided an inclusion criterion, with three [17,20,22] (30%) also including a minimum level of anxiety. This is a large improvement from Ensari et al. [9], where 11 of 36 studies (30%) did not provide inclusion criteria at all, let alone one that stipulated the anxiety levels. All studies with eligibility criteria requiring participants to have elevated anxiety at baseline were successful, with levels being above the average score for the given scale. By providing an inclusion criterion with a given anxiety level, we can gain a further understanding of the differential effects of exercise upon different anxiety levels, as differing baseline levels of anxiety can influence outcomes. This is shown in the findings of Morais et al. [23], where exercise was only beneficial to those with high trait anxiety at baseline. The use of high-trait anxious participants was encouraged by Ensari et al. [9], and hopefully, the progress shown in the present review continues.

Only a third of the present studies (*N* = 3/9, 33%) were penalised for differing in key variables between the participant groups at baseline [16,17,20]. This is an improvement from Ensari et al. [9], in which 47% (*N* = 17/36) of the studies were penalised for this. Homogeneity among groups at baseline further allows for confidence that the baseline variables did not influence the outcomes recorded. Participants in Herzog et al. [17] differed in the reported positive and negative affect scales [45], which they reported moderated the effect of exercise on anxiety. Again, while this is an improvement from Ensari et al. [9], we are hopeful that this progress continues.

While positive progress has been made, some areas of study quality remain dissatisfactory. No studies attempted to blind on any level, although we recognise this is particularly challenging given the methods required. Consequently, the relative need to ensure other areas of study quality are adhered to may be increasingly pertinent.

Aside from blinding and homogeneity at baseline, the least satisfied items were the random assignment of participants and concealed allocation at recruitment. Concealment of allocation was satisfied five out of nine times (55%), while Ensari et al. [9] reported that no studies except one satisfied the concealed allocation criteria. However, although no studies in Ensari et al. [9] were penalised in their scoring for non-random allocation of a study group or condition order, two studies were not awarded this point in the present review. Importantly, this was not possible in Morais et al. [23], as the intensity of the second condition (treadmill) was determined by the intensity of the first condition (exergame). One other study did not specify the randomisation procedures [24]. The random and concealed allocation of study participants increases the robustness of the findings, as otherwise, there is a risk of bias influencing the outcomes of one group over another.

The nature of the control group may further influence the potential relative effect of exercise. Herzog et al. [17] discussed how participants in inactive control groups (e.g., quiet rest) may display increased anxiety due to the capacity for participants to ruminate on their thoughts, although further investigation is required to confirm this. However, this is likely dependent on the associated distraction and/or mental energy of each condition. For example, a control condition such as documentary watching (e.g., [17]) may distract and entertain the participant, while studying (e.g., [22]) may draw attention to life stressors, raising anxiety. Indeed, in Herzog et al. [17], documentary watching was found to reduce anxiety, while studying in Meier and Welch [22] showed no change. Interestingly, in one of the more successful exercise conditions, LeBouthillier and Asmundson [18] deliberately employed an ‘active’ stretching control group based on this rationale and found that exercise improved anxiety beyond that of the control group. Active control conditions account for the positive effect that movement may have while also providing some mental engagement to prevent wandering thoughts or focus on daily stressors. The use of an active control group thereby increases the robustness of their findings and provides a sample of an active control paradigm that future research may choose to employ.

Understanding the underlying mechanisms of the anxiolytic effect of acute exercise has been an ongoing challenge since Petruzzello et al. [8] suggested this should be an area of research. Researchers have examined a variety of potential mechanisms, including changes in anxiety sensitivity [36], and further hypotheses include adaptations within neuroendocrine systems, the production of endorphins, and improved psychological measures such as self-efficacy (see [32,36] for reviews). Further research into these mechanisms will inform why a variety of exercise stimuli seem to be effective and why there may not be a need to search for an optimal stimulus or exercise protocol.

This review is limited, primarily regarding the inclusion criteria. The age limit for the study participants (45 years) was relatively low, and those with medical conditions were excluded. Neither of these exclusion criteria was included in Ensari et al. [9], which does not specify an age range and includes studies conducted in groups containing medical conditions. The mean (SD) age of the participants in the studies examined by Ensari et al. [9] ranges from 15.4 (2.06) [46] to 55.3 (22.3) [47]. Furthermore, while attrition is not a relevant measure in acute bouts of exercise, this review does not consider the number of individuals who failed to complete the exercise bout(s), which may indicate feasibility or acceptability for future research. Finally, this current review is further limited by only including studies published in English.

## 5. Conclusions

More than half of the exercise conditions examined reduced anxiety, and study quality overall has improved over the eight years since the last review [9]. However, methodological limitations, such as convenience sampling, remain. We also continue to grapple with decades-old questions, first considered by Petruzzello et al. [8], such as consideration of the mechanisms influencing the exercise–anxiety relationship and the unknown role of muscle-strengthening modalities in reducing anxiety. These should be the focus of future research, along with understanding the effects of self-selected intensities. Furthermore, researchers should endeavour to ensure the consideration of baseline anxiety levels within participants in their recruitment and analysis. However, the results of this review indicate the potential for acute exercise to be recommended as a strategy for individuals to manage state anxiety. Positively, the fact that different combinations of exercise duration and intensity could reduce state anxiety means there is likely no singular optimal stimulus, and any exercise is better than none. This means that practitioners can be confident in recommending that individuals use whatever form of exercise they prefer to manage their state anxiety levels.

## Figures and Tables

**Figure 1 sports-11-00145-f001:**
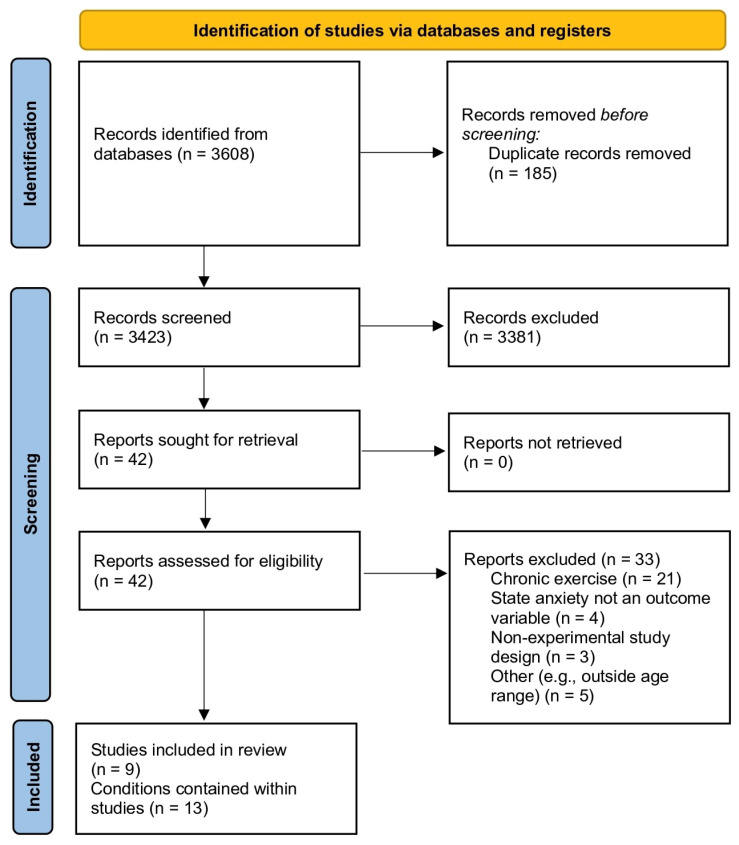
Identification of studies.

**Table 1 sports-11-00145-t001:** Characteristics of eligible studies.

Author, Date	n	Primary Intervention Time/Modality/Intensity	Control Condition(s)	Comparison Condition(s)	Measurement Scale & Mean (SD) Baseline Anxiety Score	Assessment Points	Main Result	Effect Size
LeBouthillier and Asmundson, 2015 [18]	41	20 min/spin bike sprint interval/vigorous intensity @ 60–80% age-adjusted heart rate reserve	Stretching		ASI-3 17.5 (13.0) Exercise 28.2 (12.8) Stretching	Pre, immediately post, 3-, and 7-days post	Anxiety significantly lower than control post-exercise, maintained across follow-up timepoints.	Exercise group: *d* = 0.70 (medium)
Broman-Fulks, Kelso and Zawilinski, 2015 [21]	77	20 min/treadmill/moderate @ 65–75 HRmax	Quiet rest	Time-matched resistance training to failure	STAI-S 30.3 (6.7) Aerobic 30.7 (6.3) Resistance 28.5 (7.4) Control	Pre, 5 min post	No conditions significantly reduced anxiety.	No change in any condition
Meier and Welch, 2016 [22]	32	10 min/treadmill/self-paced intensity	Biofeedback breathing task OR Studying		STAI-T: 25.9 (8.5) PSS: 22.1 (4.3)	Pre, immediately post, and 15 min post	Anxiety significantly reduced post biofeedback condition; no change with other conditions.	Biofeedback condition: η^2^ = 0.27 (large) Interaction effect: η^2^ = 0.12 (large)
Cooper and Tomporowski, 2017 [20]	64	25 min/cycle ergometry/moderate intensity @ 45% V̇O2 reserve	Quiet rest		STAI-T 38.5 (10.1) Exercising 40.6 (14.1) Control	Pre, 5, and 20 min post	Anxiety was significantly lower following exercise for the high-trait anxious group only.	High STAI-T: η^2^p = 0.09 (medium)
Lago et al., 2018 [16]	17	30 min/cycle ergometry/vigorous @ 60–70% heart rate reserve	Cycling @ 10–20%HRR		STAI-S Values not given	Pre, immediately post	Neither exercise condition significantly reduced anxiety	Insufficient data
Mason and Asmundson, 2018 [19]	63	45 min/cycle ergometry/moderate intensity @ 70% age-predicted HRmax	Waitlist	Sprint intervals/vigorous @ 85%HRmax	ASI-3 19.6 (14.6) WL 20.8 (15.7) MICT 22.4 (15.9) SIT	Pre, post, 3-, and 7-days post	Significant reductions with both exercise conditions compared to control group. Reductions were maintained in both groups at both follow-up points.	Moderate intensity *d* = −0.45 (small) Sprint intervals: *d* = −0.35 (small)
de Oliveira et al., 2021 [24]	60	30 min/single player volleyball exergame @ HR of 116–119(17–18) beats per min		multiplayer mode	STAI-T 38(13)—single player 35(13)—multiplayer	Pre, post	No anxiety reduction following either condition.	Single player: rB = 0.14 (small) Multiplayer: rB = 0.43 (medium)
Morais et al., 2021 [23]	20	45 min/dance exergame/vigorous intensity @ 75(9.8)% Hrmax		Intensity-matched treadmill	STAI-T 44.8(8)	Pre, immediately post, 10 min post	Anxiety significantly reduced in both conditions at 10 min post; exergame sig lower immediately post.	Time effect: *d* = 1.29 (large)
Herzog et al., 2022 [17]	88	30 min/cycle/moderate intensity @ 60–70% Hrmax	Watched nature documentary		STAI-S: 37.3(10.43)	Pre and post	Anxiety significantly reduced in both control and exercise conditions.	Insufficient data

Note: HR—Heart rate; V̇O_2_—Volume of oxygen uptake; STAI—State-Trait Anxiety Inventory; ASI-3—Anxiety Sensitivity Index; PSS—Perceived Stress Scale; *d*—Cohen’s d; η^2^—Eta squared; η^2^p—Partial eta squared; rB—rank-biserial correlation (non-parametric version of Pearson’s correlation coefficient). Assessment of outcomes.

**Table 2 sports-11-00145-t002:** Evaluation of study quality.

Paper	Eligibility Criteria Specified •	Subjects Randomly Allocated	Concealed Allocation	Groups Similar at Baseline	Blinding of Subjects	Blinding of Therapist	Blinding of Assessors	Measurements Obtained from than 85%	All Subjects Completed the Treatment or Control as Allocated, or Intention to Treat	Results between Groups Are Reported	Both Point Measurements and Measurements of Variability	Sum
LeBouthillier and Asmundson, 2015 [18]	1	1	1	1	0	0	0	1	1	1	1	7
Broman-Fulks, et al., 2015 [21]	1	1	1	1	0	0	0	1	1	1	1	7
Meier and Welch, 2016 [22]	1 ^	1 °~	0	1	0	0	0	1	0	1	1	4
Cooper and Tomporowski, 2017 [20]	1 ^	1	1	0	0	0	0	1	1	1	1	6
Mason and Asmundson, 2018 [19]	1	1	1	1	0	0	0	1	1	1	1	7
Lago et al., 2018 [16]	1	1 °~	0	0	0	0	0	1	1	1	1	4
de Oliveira et al., 2021 [24]	1	0	1	1	0	0	0	1	1	1	1	6
Morais et al., 2021 [23]	1	0 °	0	1	0	0	0	1	1	1	1	5
Herzog et al., 2022 [17]	1 ^	1	0	0	0	0	0	1	1	1	1	5

Note: ^ indicates anxiety level specified within inclusion criteria, ~ indicates counterbalancing, ° indicates cross-over design, • item is not included in sum calculation as per instrument instructions [8].

## Data Availability

No new data were created or analysed in this study. Data sharing is not applicable to this article.

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
