# Peer review of "The Effect of Acute Exercise on State Anxiety: A Systematic Review"

_sports, 2023, doi:10.3390/sports11080145_

Round 1

Reviewer 1 Report

This manuscript has no particular problems, but the biggest drawback is that it does not convey the inevitability or necessity of conducting a review of acute exercise and state anxiety at this time. I think it is necessary to emphasize in the introduction. Also, although it is a minor point, there is a sentence that does not make sense on the 5th line from the bottom of page 3, so please change it appropriately.

Reviewer 2 Report

Thank You for Your excellent paper

. I coudln't find any week points in Your research strategy and describing results.

I like specially Your discussion on potential influential factors and explanations of observed results.

I believe, researchers can be inspired in next few years by Your work.

Reviewer 3 Report

This is a very well written and detailed review of studies examining a potential effect of acute exercise on state anxiety. The manuscript is well structured and the descriptions of conditions used in the reviewed studies are comprehensive.

I don’t have any major concerns: however, I was surprised to see that the authors did not report effect sizes in Table 1. The findings of the studies are mixed as it is (with 53% studies showing the effect, the remaining not), so the information about effect size in the ones that did show the effect would be very informative (and since studies are published after 2015., I am confident that the authors had to report the effect size, or at least Ms and SDs from which it can be calculated).

Furthermore, I would like to see a more comprehensive conclusion: the authors provided some valuable suggestions regarding the future studies, but what would be take home message from the studies reviewed in this paper? What would the authors suggest to, say clinician, who is thinking about implementing this sort of intervention in their work with patients suffering from anxiety? (again, this is related to my previous comment, as the information about the effect sizes could be important when making his decision)

Minor comments:

-          First sentence in study selection: “As detailed in Error! Not a valid bookmark self-reference…

-          References – something happened with the font (it is barely readable)

Round 2

Reviewer 1 Report

The manuscript has been appropriately revised to make the author's conclusions more persuasive, and I believe it is acceptable for publication. However, I still have the impression that the research period was short and the number of research papers was too small.